# Estimating global bee species richness and taxonomic gaps

**James B. Dorey** [1,2,3] ✉, **Amy-Marie Gilpin**[4,5], **Nikolas P. Johnston**[1,2], **Damien Esquerré**[1,2], **Alice C. Hughes**[6], **John S. Ascher**[7,8] & **Michael C. Orr** [9,10,11] ✉

Robust species richness estimates are critical for meaningful conservation prioritisation, understanding ecosystem resilience, and studying evolutionary processes. Yet, they remain elusive even for some of the best-studied groups and regions. As keystone pollinators, bees are crucial for the maintenance of healthy ecosystems, yet rigorous estimates of their species richness are lacking. Here, we statistically estimate global, continental, and country lower bounds of bee species richness. Globally, we estimate 24,705–26,164 bee species, an 18–25% increase, representing at least 32–45 years of taxonomic research. We estimate particularly high undescribed biodiversity from Asia, Africa, and the Americas. We find that taxonomic gaps are correlated with gross domestic product per capita (GDPc), observed species richness, number of occurrence records, and completeness of occurrence databases. Our statistical R-package framework will progress our understanding of lesser-known groups, downstream consensus, and mobilise existing occurrence datasets to quantitatively estimate species richness on global scales.

Quantifying species richness is a fundamental aim of biology. Species richness influences ecosystem resilience[1], conservation[2], policy[3], and our understanding of evolutionary processes[4]. Reliable richness quantification is difficult, and our inability to do so represents 'a colossal failure of science'[5], with global species richness estimates varying wildly from 2 million[6] to 1 trillion[7]. Currently there are ~2.2 million described species on Earth[8] with global rates of eukaryotic species description at about 18,000 species/year; 75% of which are invertebrates[9]. However, rates of species description have varied across time and taxa[10]. Understanding species richness is non-trivial and has considerable real-world implications for species conservation in the face of anthropogenic threats[11], placing taxonomy as the cornerstone of conservation assessments and biological research[12]. There

is a bias in species descriptions to large and charismatic organisms. For instance, it is generally accepted that relatively few new bird species remain to be discovered[13], and this holds for most other vertebrate groups as well (however, see bats[14]). But outside of these well-studied groups, even critical ecosystem engineers, such as trees, have an estimated 9000 undescribed species[15], in addition to the described 64,088 species (+14%)[15]. Estimates of undescribed species, such as this, can influence and inform the allocation of funding, research, management, and conservation efforts.

The vast majority of described animal species are insects[16]. Insects are so overwhelmingly diverse that May[17] noted, 'To a good approximation, all species are insects'. Estimations of this diversity range from 'guesstimates' to more realistic data-driven estimates; there is much

[1]Environmental Futures Research Centre, School of Science, University of Wollongong, Wollongong, NSW, Australia. [2]Molecular Horizons Research Institute, School of Science, University of Wollongong, Wollongong, NSW, Australia. [3]College of Science and Engineering, Flinders University, Bedford Park, SA, Australia. [4]School of Science, Western Sydney University, Penrith, NSW, Australia. [5]Hawkesbury Institute for the Environment, Western Sydney University, Penrith, NSW, Australia. [6]School of BioSciences, Faculty of Science, University of Melbourne, Melbourne, VIC, Australia. [7]Department of Biological Sciences, National University of Singapore, Singapore, Singapore. [8]Lee Kong Chian Natural History Museum, Singapore, Singapore. [9]Entomologie, Staatliches Museum für Naturkunde Stuttgart, Stuttgart, Germany. [10]Institute of Zoology, Chinese Academy of Sciences, Beijing, China. [11]KomBioTa-Center for Biodiversity and Integrative Taxonomy, University of Hohenheim and State Museum of Natural History, Stuttgart, Germany. ✉e-mail: jbdorey@me.com; michael.christopher.orr@gmail.com

uncertainty across the diversity of applied methods, but reliable approximates range from 2.6 to 7.8 million insect species[16]. The lack of consensus on the number of insect species is especially concerning given the reported widespread decline in insects[18,19] and the low number of species currently described (one million or 13–38% of expected diversity)[20]. Advances in molecular techniques allow cryptic taxon discovery and robust taxonomic hypotheses; however, validating names and integrating molecular and morphological evidence (via integrative systematics workflows) remains challenging[21]. Even for some of the best-understood insect taxa, global or regional richness estimates are rare. For example, a recent large-scale collaborative work on ants focused on discovering the distribution of unknown species; rather than explicit quantification of richness[19]. However, this method provided a useful 'treasure map' for targeted species discovery. Despite their status as the pre-eminent animal pollinators[22], to our knowledge, there are no statistical estimates of bee (Anthophila) species richness, with haphazard species richness estimates mis-cited and proliferated in literature and science communication. In 2007, Michener[23] estimated that there were ~18,000 described species with >20,000 total in the world. The Discover Life bee taxonomy contains ~21,000 valid species[24,25], globally; yet many new species are hypothesised to exist, especially in Australia, China, and Argentina and Chile[26]. Recent bee revisions have seen increases of up to 96%[27] and, through synonyms, reductions of 57%[28] of species in other groups. However, most bee groups have not been subjected to in-depth taxonomic assessment, and thus, we have no clear idea of the total number of bee species.

Here, we generate statistically derived and quantitative global-, continental-, and country-level estimates of bee species richness. We combine a global bee occurrence dataset (8.3 million cleaned records), taxonomy (~21 k valid names and ~24 k synonyms), country checklist (~66 k species-country pairs, including almost all valid names)[24,29], and literature records (497 species), with statistical estimators of species richness (iChao1 and Hill numbers). We additionally propose a major hypothesis and several predictions to explain patterns in our estimated taxonomic gaps.

We hypothesise that contemporary and historical differences in socioeconomic and scientific investment as well as physical regional characteristics have generated uneven rates of bee species description across regions. We propose several predictions in relation to physical and social diversity gap correlates and our response variables, predicted number and percentage of new species, to address this hypothesis. First, due to increased niche diversity and potential difficulty in sampling, we predict that our response variables will be positively correlated with elevational range and country area. Second, because increased median distance to nearest road could make sampling for bees more difficult, our response variables will be positively correlated with median distance to nearest road. Third, we predict that our response variables will be positively correlated with observed species richness. This is because we expect that the taxonomic load might be higher; however, we also recognised that it could indicate that a larger portion of the bee diversity was already described. Fourth, we predict that our response variables will be negatively correlated with gross domestic product per capita (GDPc) and education. This is because we expected that GDPc and education to be indicators of a country's investment into research and a greater capacity to undertake taxonomic works locally or to host collaborating taxonomists. Finally, we predict that, due to greater inventory and taxonomic efforts, our response variables would be negatively correlated with the number of clean occurrence records and the proportion of occurrence-based species per country.

The framework and R package (BeeBDC ≥v1.3.1[25]) toolset we provide can be used for estimating actual taxonomic diversity across larger taxonomic and geographic scales.

## Results and discussion
### Global, continental, and country bee diversity

We statistically quantified the number of bee species at the global-, continental-, and country-levels for 186 countries and provided custom statistical tools for use with other taxa (BeeBDC v1.3.1[25]; Supp. Materials). Over 100 iterations, we estimated the lower bound of bee species richness globally to be between 24,705 and 26,164 species (lower and upper 95% confidence intervals; Fig. 1A). This equates to an estimated taxonomic gap of 3771–5230 species, or an 18–25% increase on the 20,934 currently recognised bee species[24,25]. With the exception of our global (iNEXT, richness rarefaction and extrapolation, 24,985; 24,705–25,266; +16% and iChao, non-parametric richness estimation, 25,929; 25,705–26,164; +19%) and Asian (iNEXT 8275; 8057–8473; +34% and iChao 8708; 8528–8902; +41%) median estimates, all 95% confidence intervals of both estimate methods (iNEXT and iChao) overlapped, providing greater confidence in our estimates (Figs. 1A–C and S1–S4). Because of this, and for simplicity, we report on iChao below.

At the continental level, Asia, including the Middle East, had the largest taxonomic gap of 2525 (+40%) species; although, in terms of percentages, Africa (1668; +34%), Europe (568; +27%), Oceania (472; +23%), and South America (1262; +29%) also exhibit large gaps (Figs. 1B, 2, and S5, S6). The gap for North America (1010; +18%), while lower as a percentage, is still large and represents years of taxonomic research at present global discovery rates (Figs. 1B, 2, and S5, S6; see below). Furthermore, at the continent and country levels where land boundaries are shared, estimates will include described species that could occur in neighbouring regions.

Our estimates provide a powerful guide and useful hypotheses for where to concentrate species discovery and revisionary research. For example, we estimated a taxonomic gap of 843 (+46%) species for Turkey, which is larger than estimates for the entirety of continental Europe. Of the remaining top ten most species rich countries (Figs. 1C and S7, S8), China (637; +47%) and Israel (553; +52%) have particularly high taxonomic gaps. Many regions identified here with high taxonomic gaps have been modelled, based on environmental factors (especially xeric areas, solar radiation and non-forest plant productivity), to have high species richness, including Turkey, China, and Argentina[26]. However, we also found discordance with countries like Australia, Chile, and Madagascar showing relatively low estimated undescribed species diversity (Fig. 2C) but otherwise high modelled richness[26]. This discordance doesn't appear related to recent descriptions in those regions and should be the topic of further, potentially molecular, research.

Our estimates are likely conservative as they estimate the lower bound of richness and taxonomic revisions are frequently undertaken without the inclusion of molecular techniques. For example, even in a country with high GDPc like Australia, only 12% of new species described since 2000 were named in articles that formally used phylogenetic analyses (Table S1). In a recent and contrasting example, the morphologically cryptic Fijian *Lasioglossum* bees went from four morphologically described species to 13 when systematic techniques were included[30], and recent molecular evidence suggests at least 28 total species[31]; a 3.3- and 7-fold increase, respectively. Hence, we should expect that these estimates will increase overall as molecular sampling techniques are implemented more broadly, sequencing costs fall, and data availability increases[32]. However, it is important that these techniques are used reliably and without over-interpretation[32,33]. Additionally, sequencing efforts can, and do, result in the discovery of synonyms (rejections of existing species hypotheses)[34]. Large barcoding efforts may elucidate, by region, the extent to which existing taxonomies have over- or under-described species.

Wild bee conservation, research, and taxonomy around the world faces challenges, including regionally poor data, extinctions, poaching,

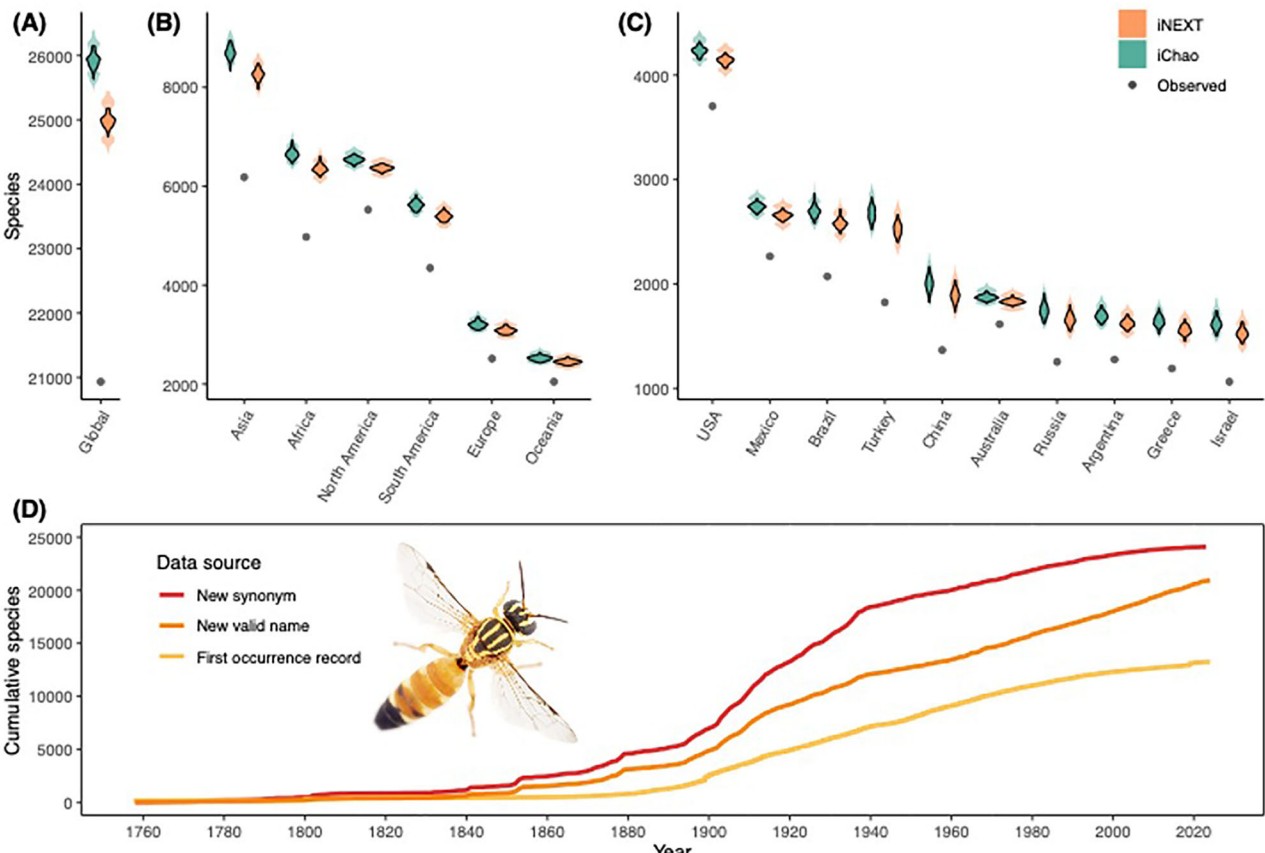

**Fig. 1 | Observed and predicted bee species richness.** The (**A**) global, (**B**) continental, and (**C**) top and bottom ten countries showing the number of observed species (grey) and the median number of estimated species using the statistics iChao (green) and iNEXT (orange) after 100 iterations of sampling. The median 95% confidence intervals are shown in the shaded violin plots. The (**D**) accumulation curves of all new synonymous names (red), new names that remain valid today (orange), and the first publicly available occurrence record (yellow) for all bee species in the world. The bee image is a *Palaeorhiza* species photographed by James Dorey. Source data are provided as a Source Data file.

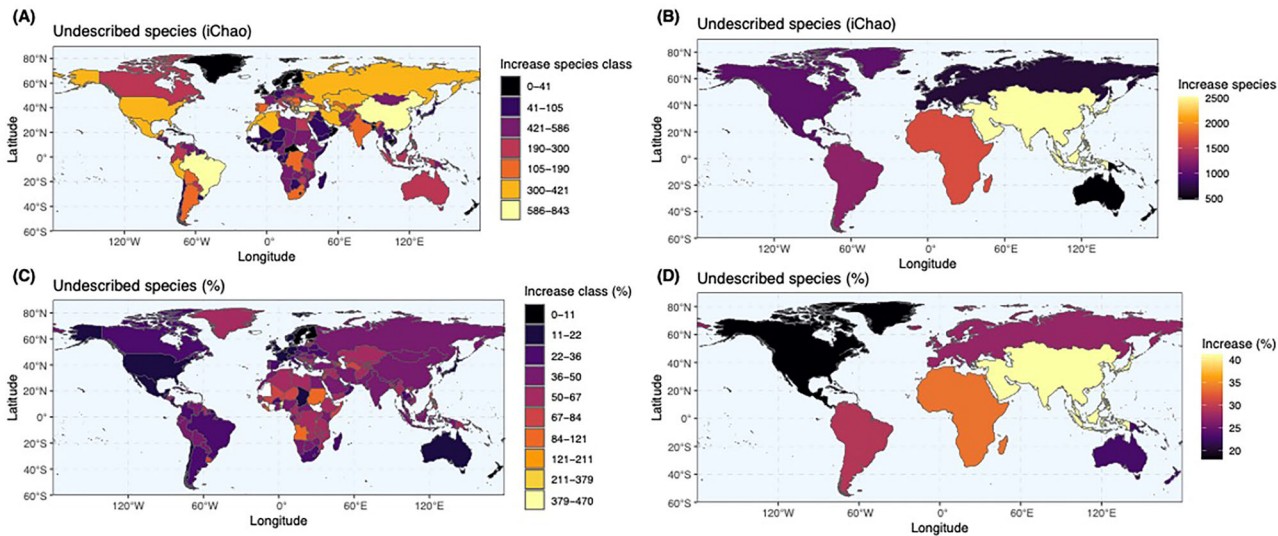

**Fig. 2 | Maps of the estimated (iChao) number and percentage of species by country and continent.** The estimated number and percentage, respectively, of undescribed species broken into classes by (**A**–**C**) country and displayed continuously increases by (**B**–**D**) continent. Estimates are from the median values of 100 iterations sampling the literature curve combined with the empirical data. Lighter colours indicate higher values. Base maps were sourced using rnaturalearth version 1.0.1 and rnaturalearthdata version 1.0.0[51,80]. Source data are provided as a Source Data file.

rare conservation assessments, political and cultural barriers, expert deficits, museum backlogs, and national restrictions to sharing and collaboration[26,35–39]. These challenges are clear in Asia, which has the highest observed and predicted species richness and taxonomic gap (Fig. 2 and S5), but major issues with taxonomy and species discovery capacity[40]. Issues are also apparent for Africa, which potentially lacks ample data (Figs. 2 and S6) when compared to Orr et al.[26], who suggested high relative richness in North, East, and South Africa. Chad is the closest African country to asymptote with a predicted 17% taxonomic gap (globally 20th smallest gap for $n \geq 300$; while 14/20 are European). Many Central and South American countries are also far from asymptote (Fig. S4). However, continentally, there are trends towards asymptote (Fig. S3). In contrast, Europe has relatively few remaining species to describe with countries like Sweden and Switzerland nearly at asymptote (Fig. S4). The low predicted taxonomic gap in Oceania was unexpected and disagrees with recent qualitative analyses[26]. However, Australia holds 94% of Oceania's clean–filtered and reliable[29]–bee occurrence records with few integrative systematic descriptions (e.g. Table S1). This likely results in cryptic diversity going unrecognised and leading to a scarcity of singleton and doubleton species, which would drive the lower regional diversity estimates. This, in combination with a lack of recent and thorough integrative systematic research in Oceania, could explain the lower estimates where we otherwise expected higher richness. A concurrently published paper, not included in our analyses, highlighted this issue by describing 71 new species of Australian *Megachile (Austrochile)* (28% of the estimated taxonomic gap of 250 species) using systematic techniques[41]. It might also be a broader issue for countries where molecular techniques are inadequately implemented in taxonomic research; such countries are likely the majority, and most new arthropods are described using morphology alone[42]. However, non-statistical estimates of species richness in Australia–'as high as 2000'[43], 1947–2047[44], ~2600[45], and 2–3000[46]–do not overlap with, but are close to, our estimate of the lower bound for Australia, 1759–1919 (Fig. S4).

Our work also provides insights into how bee species richness varies between continental and island countries. Previous modelling could not effectively estimate even relative island species richness[26]. Our approach better incorporates biogeographic limitations to estimate that species richness for island nations is significantly higher ($p = 5.0 \times 10^{-8}$), per unit area, than continental nations (Fig. S9). Isolated islands are known to host large numbers of endemic species and simultaneously be disproportionately threatened by anthropogenic changes[47]. Hence, we highlight that islands should be a priority for taxonomic and conservation research.

The development of integrated, novel, and accessible workflows, like ours, to pinpoint discovery and collaboration opportunities, help allocate efforts and prioritise funding accordingly, is a major advance. Our workflow quantitatively, rather than relatively, estimates richness at multiple macro-scales and is applicable to any taxon and region with sufficient data. Our taxonomic gap analysis quantifies global biodiversity patterns and highlights the urgent need for investment in the global south and parts of the global north.

## Species accumulation and discovery rate

Since 1960, new bee species names that remain valid today have accumulated at the relatively constant rate of 117 species/year (Fig. 1D). This likely indicates that the rate of description is limited by a taxonomic bottleneck rather than availability of new species. Applying this rate directly, we might expect to close the taxonomic gap in 32–45 years. However, given that our richness estimates are a lower bound, the integration of phylogenetic analyses can be very limited (Table S1), research funding and capacity are geographically biased[40], reduced support for taxonomy, and the likely diminishing returns of species revision towards the tail end of discovery, we expect this estimate to be quite conservative.

Synonymy of more-recent names has slowed since about 2000, while the rate of new valid names has not (Fig. 1D). This could indicate that the reliability of bee taxonomy is increasing overall but that we are not starting to exhaust the number of new species left to describe. Additionally, from the 497 species for which we collected sample size data from the most-recently published revisions, 226 (45%) were described from a single specimen (Fig. S10), further supporting the notion that there are many remaining bee species to describe. In a similar timeframe, the number of first digitised occurrence records for each species has also fallen behind the rate of new valid names (Fig. 1D). This, along with our literature search (Supp. Materials), indicates issues in the open sharing of occurrences with taxonomic works and a shortfall in museum digitisation efforts. Even in developed countries, occurrence record shortfalls are a major issue for conservation assessments as well as species discovery[32,38]. These issues are likely broadly applicable to many other taxa. Removing barriers to uploading research data and enforcing open science practices will be critical to addressing these issues in the future. Furthermore, standardised approaches to digitising and sharing such data in ways that are easy to aggregate (e.g. using the Darwin Core occurrence data standard–dwc.tdwg.org–which is required by some journals such as the European Journal of Taxonomy) would greatly facilitate our ability to collate these data and track species descriptions.

## Diversity gap correlates

To examine the factors contributing to taxonomic shortfalls, we used linear mixed-effects models with countries as samples and continents as the random effect. We analysed correlates of the number and percentage of new species. Our correlates included GDPc[48], tertiary education[49], the observed species richness, the number of BeeBDC-cleaned bee occurrence records[29], elevational range[50], country area[51], country-wide median distance from roads[52], and the proportion of occurrence-based species versus species drawn from the literature curve. We consistently found no significant relationship between our response variables and the percentage of tertiary education, median distance to roads, elevational range, or country area (Tables S2 and S3).

In agreement with our prediction, increasing GDPc was negatively associated with the estimated number and percentage of undescribed species (Figs. S11A and S12A). This was expected because richer countries, in general, should have described a greater number and proportion of their bee fauna due to greater investment in science and research capacity. However, for Africa, increasing GDPc appeared to be associated with an increase in the number of undescribed species (Fig. S11A). This likely reflects an overall larger taxonomic gap for Africa and an inability to reliably estimate trends due to a lack of data (Fig. S11F, G). In such cases, it seems likely that even initial efforts at collections and taxonomic research in the region will uncover many new species and greatly enhance our ability to work with and understand the fauna.

Observed species richness had a very strong, consistent, and positive impact on the estimated number of new species in every continent (Fig. S11F). This likely indicates that bee-diverse regions are expected to have a larger number of species remaining undescribed, supporting our prediction for this variable. Even in the best studied regions, such as North America, many species may persist undescribed in larger groups (such as *Andrena*, *Lasioglossum*, or *Osmia* in the USA) that are best treated altogether to enhance identifiability and ensure accurate taxonomic actions. In contrast, the positive correlation between the percentage of new species and observed species richness appears to mostly be driven by Asia, with wide confidence intervals for other continents (Fig. S12F). This reinforces the suggestion that Asia, including the Middle East, is a standout priority in terms of addressing the taxonomic gap for bees[40].

Both the number of clean occurrence records and the proportion of occurrence-based species appeared to have a negative association

with both the number and percent of undescribed species (Figs. S11G, H and S12G, H) and both had a weak but significant interaction (Tables S2 and S3). However, our model indicated that the proportion of occurrence-based species had a positive association with both response variables (Tables S2 and S3). Without log-transformation, we supported our prediction that the number of clean occurrence records and the proportion of occurrence-based species were negatively associated with our response variables because both variables indicate greater sample completeness associated with more taxonomic work and material (Supplementary Materials). This makes intuitive sense, as areas with more data should have undergone more taxonomic research and identification work necessary to generate such data. Across regions, we see great variability in taxonomic completeness and gaps. There are opportunities for foundational and traditional taxonomy in areas such as Africa to more coordinated efforts on larger or more challenging groups in places such as North America. For the former, integrative systematics will build strong taxonomic scaffolds early, and for the latter, it will allow robust testing of existing species hypotheses.

Hence, GDPc, observed species richness, the number of clean records, and the taxonomic completeness of digitised occurrence records are good indicators of the taxonomic gap. Not only are the latter three variables the strongest correlates, but they are also the ones that can be improved through building capacity and collaborations. Our results support the socioeconomic component of our hypothesis in generating uneven rates of bee species description across regions. However, we reject the physical regional characteristic component for the variables that we examined.

To determine the potential impact of sampling species counts from the empirically derived literature curve, we further sampled two curves based on the empirical country- and global-level occurrence datasets. Most estimates were robust to changes in the chosen curve, with the literature-sampled occurrence curve sometimes producing higher estimates than the global or country curves (Figs. S10 and S13). Except for GDPc, and—for the global dataset—the interaction term between the number of clean records and the proportion of occurrence-based species, the significance and sign of the remaining fixed effects were consistent (Table S3). This suggests that our use of the empirical literature curve is a robust choice and hence useful for implementation with other taxa.

We applied several tests of the data and found that the significant patterns and directions of fixed effects in our linear mixed-effects model were mostly robust to changes in the models; however, changing the data transformations could alter the results. Importantly, when the number of clean records and the proportion of occurrence-based species were not log-transformed, the significance values were identical; however, both fixed effect variables exhibited negative associations with the response variables.

Further diversity gap correlates could be explored if those data were reliably compiled. Phylogenetic signal is one possibility, and a global super-matrix bee phylogeny exists[53]; however, the available data for some groups are insufficient to make reliable inferences. While we try to understand taxonomic capacity through variables such as GDPc and tertiary education, datasets that describe the origin and employment locality for taxonomists might also explain some of our patterns. However, taxonomists do not solely work on taxa from those places. Verification of our results could also be achieved by surveying the number of undescribed species in museums. However, this would require a colossal digitisation effort, which has not been adequately completed even for described materials or taxa.

## Broader implementation

For decades, we have been unable to estimate species richness or its uncertainty, at least partly because easily-implemented methods did not exist[54]. This explains the rarity of global quantitative species richness estimates. However, global occurrence, taxonomic, and checklist datasets are quickly growing in size and completeness. For example, the Global Biodiversity Information Facility has over three billion occurrence records currently listed[55]. Other repositories[29], data-providers (e.g. iNaturalist)[56], and taxonomic/checklist providers[57] similarly continue to appear and grow. However, different datasets should be judged and used based on their relative merits and reliability for each taxon of interest[58,59]. The tools for reliably dealing with these occurrence data, identifying synonyms, and cleaning data are also becoming rapidly more available and increasingly comprehensive[25,60,61]. Bees are not the only understudied taxon for which sizeable datasets exist. Given the availability of data and processing pipelines—e.g. integrated species occurrence data cleaning and species richness estimation functions now in BeeBDC[25]—these kinds of analyses can be implemented for other taxa with relative ease. Such research will provide quantitative, statistical, and geographically explicit roadmaps for prioritising the description of whole taxonomic groups.

Such analyses should be made and explained with caveats kept in mind. For example, while iChao and Hill numbers are relatively robust estimators of species richness within assemblages[62–64], our 'assemblages' might be considered very large. Global species occurrence datasets, while relatively extensive, have inherent biases[29]; however, biases abound for perhaps all sampling methods[65]. We expect that taxonomic concepts will also impact our estimates, particularly where rare or cryptic species are left undescribed or lumped. While we state species richness estimates here at several levels, we highlight that they are estimates of the lower bound of diversity, biases are likely to further decrease estimates[62–64], and that estimates can be somewhat sensitive to input data (e.g. Figs. S10 and S13). We suggest that our estimates, while far better than haphazard estimates, are appropriately used as statistically derived hypotheses and roadmaps for future work and management.

## Summary

Understanding the number of species on Earth remains an enduring and unanswered question in biology. We provide quantitative global-, continental-, and country-level estimates for all bee species. Our prediction of bee richness globally is 24,705–26,164 species, which represents a lower-bound. We conservatively estimate 32–45 years of taxonomic work, which is certainly an underestimate and unevenly distributed across the globe, with most new bee richness expected in Asia, followed by Africa, South America, North America, Europe, and then Oceania. The relatively low estimate for Oceania was unanticipated. However, we caution that this might be driven by relatively high data availability in Australia, but poor regional availability, and insufficient use of molecular techniques, even in recent taxonomic research. Insufficient implementation of integrative systematics is an issue for many countries and taxa. Our analyses also highlight that, for bees and per unit area, island nations are more species-rich than continental countries, indicating the evolutionary and conservation importance of islands beyond simple endemism.

We provide the methods needed to prioritise bee species discovery and taxonomic investment (see vignettes https://jbdorey. github.io/BeeBDC/). We suggest that countries with lower research capacity in taxonomy (e.g. fewer taxonomists) need to be prioritised, and that wealthy countries should aim to build regional capacity and collaborations to address shortfalls. We further suggest that molecular techniques should be used whenever possible in combination with morphometric data (integrative systematics) to correctly identify and describe cryptic diversity, build robust species concepts[66], identify potential synonyms, and generate better estimations of taxonomic gaps. While there are challenges in building reliable estimates of species richness, particularly for data-poor countries, our analyses enable powerful conclusions to be drawn with real-world applications for

taxonomic and conservation prioritisation. With ever-growing occurrence datasets, checklists, and accessible programming tools, our research also provides a statistical and computational template to estimate species richness across different taxa. With broader adoption and expansion of these methods, we hope to address the 'colossal failure of science'[5] that is our inability to robustly quantify species richness at large geographic and taxonomic scales.

## Methods

### Data acquisition and manipulation

We used the BeeBDC species occurrence data and the Discover Life taxonomy and country checklist downloaded in BeeBDC version 1.2.0[24,25,29]. We filtered the uncleaned version of this dataset to remove records with (i) invalid binomials, (ii) improper basis of record, (iii) unmatched coordinates and country name, (iv) absences, (v) invalid license, or (vi) outside of their country checklist using BeeBDC[25] in R version 4.4.1[67]. We manipulated data mostly using tidyverse[68] packages and created plots using ggplot2 version 3.5.0[69] and iNEXT version 3.0.1[63]. All code for data management, manipulation, analyses, and figures is provided on our GitHub (https://github.com/jbdorey/BDE_R_wofklow) and core functions are available with BeeBDC ≥v1.3.1 (https://jbdorey.github.io/BeeBDC/).

For valid species names that lacked any occurrence records, we applied a distribution from empirical sample sizes using the most recent revisions as follows. For the 4819 species that had no occurrence records, we found sample sizes using one of two methods. Firstly, we sought complete occurrence datasets from the literature. On the 12th of April 2024, to find data from taxonomic revisions, we searched Scopus with the term 'bee AND revision AND taxonomy' and looked at all multiple species bee papers with citations >2 ($n = 53$; citations available from https://github.com/jbdorey/BDE_R_wofklow SuppMaterials/ScopusSearch). Of these papers, only one provided data in a tabular format[70]. We also extracted occurrence datasets from papers where the present authors knew that datasets were provided with the publication[4,27,38,71–75] and almost 100 additional records that were digitised (see https://github.com/jbdorey/BDE_R_wofklow/blob/main/SuppMaterials/DiversityLiteratureRecords.xlsx). Secondly, we randomly extracted 600 of the 4819 species names that lacked any occurrence records and extracted sample sizes from the most recent taxonomic revisions that covered each taxon (see https://github.com/jbdorey/BDE_R_wofklow SuppMaterials/1.6c_randomLitTogether.csv). Across these literature records, we got sample sizes for 497 or >10% of all missing species, that lacked any occurrence points in the BeeBDC dataset (see https://github.com/jbdorey/BDE_R_wofklow/blob/main/SuppMaterials/1.6c_randomLitTogether.csv).

We estimated the curve of best fit to these literature data using Mosaic version 1.9.1[76]. We used the formula $y = (226.6 \cdot x) \cdot x^{-\log(10.7)}$, which had the lowest residual standard error (6.1) that we could achieve (Fig. S10). For our global species richness analysis, we used this distribution to randomly generate sample sizes for the remaining 4322 no-occurrence species. For the global-, continent-, and country-level analyses, and for those species that occur in the Discover Life country checklist but not in the occurrence dataset, we applied the above formula and capped the distribution to the maximum empirical species sample size for each country.

To compare the rate of new species descriptions, first occurrences, and synonyms, we needed to filter orthographic variants out of the greater synonym list. To do this, we grouped our data by accepted id (i.e. accepted name) and author year (without brackets). We then removed numbers from names, and compared all within-group combinations using tidystringdist version 0.1.4[77] and igraph version 2.1.4[78] and accepted only the first of each orthographic variant. This resulted in a total of 24,166 synonyms from a starting list of 30,831. We used the year of description to build the synonym accumulation curve; hence, the rate of accumulation does not equate to the year when names were synonymized but their initial description.

### Estimates and visualisations of species richness

To examine patterns of species richness and discovery, we analysed several datasets in statistically independent ways. Firstly, we used the bee taxonomy and the year of first description to plot species accumulation (Fig. 1D). Secondly, we used the combined species occurrence and literature datasets (globally, by continent, and by country) and estimated (i) the lower bound of species richness using iChao1[62] with SpadeR version 0.1.1[79] and (ii) Hill numbers using iNext 3.0.1[63]. In order to run SpadeR and iNEXT analyses, we built functions, ChaoWrapper, iNEXTwrapper, and ggRichnessWrapper, that allow multiple levels to be analysed at once in parallel (https://github.com/jbdorey/BeeBDC). The iChao estimate is known to perform well when species abundances in 'communities' are homogenous and can be negatively biased where they are heterogenious[62]. Hill numbers, as estimated in iNEXT, are implemented to make fairer comparisons between assemblages that may vary in size[63,64]. We further built a function, richnessEstimateR, that allowed us to iteratively sample the literature curve 100 times at the global-, continent-, and country-levels and extract the estimates, upper, and lower confidence intervals for both statistics. Especially at the global level, these analyses took up to a couple of weeks, even using a 20-core MacPro. These functions are deployed in BeeBDC ≥v1.3.1.

### Country-level exclusions

We excluded countries from analyses for several data-related reasons. Firstly, we entirely excluded all countries with a sample size (number of species occurrence records) of less than 30. Secondly, we excluded countries where the species estimates were excessively different (by an order of magnitude) from the observed species richness. This was usually because the number of occurrences approximately equalled the number of species, because the maximum species-level sample size was one.

### Diversity gap correlates

We looked to determine the potential drivers of diversity gap estimates; i.e. predictors of the number or percentage of species remaining to be described per species. We correlated our iChao estimates against GDPc[48], tertiary education[49], the observed species richness, the number of clean BeeBDC records, elevational range[50], country area[51,80], median distance from roads[52], and the proportion of species derived from the literature curve. We analysed these variables using a linear mixed-effects model in R package, *lmerTest* version 3.1-3[81] and iteratively removed interaction terms. Linear mixed-effects models are versatile models that are also reasonably robust to mild violations of assumptions, such as skewed data and correlated predictors[82]. When trying to explain the (i) number of new species predicted and (ii) the predicted percentage increase of species from iChao, we took the log of all variables and included an interaction term between the number of clean records and the proportion of species derived from the literature curve. We took the same approach to examine the relationship between the number and percentage of predicted new species.

Furthermore, and in order to examine the interaction between terms, we extracted data from a curve based off of the (a) global and (b) per-country species occurrence data frequencies as we did with the literature data (Fig. S10). These data were normalised with the literature curve to work from the same scale during formula production and plotting. We ran these analyses ten times at the country level for iChao and iNEXT to examine how changing empirical data that fills in the no-occurrence species impacts the diversity gap analyses. We ran further tests by removing combinations of fixed effects, applying different

transformation factors, iteratively removing interaction factors, applying different transformations (none, log, exponential, or square root), and applying iChao- or iNext-derived sample sizes rather than the raw number of clean records.

## Reporting summary

Further information on research design is available in the Nature Portfolio Reporting Summary linked to this article.

## Data availability

Species occurrence, taxonomy, checklist and all other data are available from Flinders ROADS (FigShare; BDE_R_workflow) at https://doi.org/10.25451/flinders.21709757. Source data are provided with this paper.

## Code availability

The core functions required to complete these analyses for bees or other taxa are available with BeeBDC v.1.3.1 or higher (https://github.com/jbdorey/BeeBDC). A vignette of these functions is available at https://jbdorey.github.io/BeeBDC/articles/speciesRichness_example.html. Our R scripts are all available on GitHub (https://github.com/jbdorey/BDE_R_wofklow)[83] along with input and interim files, tables, figures and additional files referred to throughout. (https://github.com/jbdorey/BDE_R_wofklow).

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

## Acknowledgements

We thank all of the researchers, taxonomists, and citizen scientists who make excellent collections and observations, pay attention to uploading quality data, and make it publicly available. We further thank the institutions and repositories that collate and publish these data for the betterment of the field. In particular, we thank the repositories that each contribute >100,000 bee occurrence records, including iNaturalist, the USDA Agricultural Research Service, Observation.org, Sveriges Lantbruksuniversitet Artdatabanken, Natuurpunt, the USGS Native Bee Inventory and Monitoring Lab, Centre Suisse de Cartographie de la Faune, the American Museum of Natural History, the Kansas University Biodiversity Institute and Natural History Museum, Biological Records Centre, Bumblebee Conservation Trust, Adam Mickiewicz University in Poznań, Naturgucker, and the Finnish Biodiversity Information Facility. We would also like to thank Laurence Packer for discussing with us the current estimates of bee species richness, Thomas Wood for general discussions, and Michael Schwarz for his discussions on statistical methods. Funding was received from the University of Wollongong Startup Grant (JBD), CAS President's International Fellowship Initiative (2024PVC0046; MCO), and Chinese Academy of Sciences President's International Fellowship Initiative (2026PVC0122; MCO).

## Author contributions

J.B.D. conceptualised the study, produced visualisations, software, formal analyses and wrote the original draft. J.B.D. and M.C.O. designed the methodology and acquired funding. J.B.D., A.-M.G., D.E., N.J., A.C.H., J.S.A., and M.C.O. collected and curated data. All authors reviewed and edited the final version of the manuscript.

## Funding

## Competing interests

The authors declare no competing interests.
