## [Transparent Peer Review File · Nature Communications]

Estimating global bee species richness and taxonomic gaps

Corresponding Author: Dr James Dorey

Version 0:

Reviewer comments:

Reviewer #1

(Remarks to the Author)

How many bee species are there? A quantitative global estimate

In this manuscript the authors estimate the number of bee species in the world. They estimate an 18–25% increase on the known diversity, and suggest that this will require at least 32–45 years of taxonomic research in order to describe the new species..

The authors take novel approaches to estimating bee diversity and it is helpful to have such a well-known group intensively studied in this way since it provides ways for others to replicate with other groups. However, in my view the authors have attempted to include too much in this manuscript and would be better served by separating it out into possibly three manuscripts. For example the socio-economic factors contributing to lack of taxonomic effort on bees should be in a separate paper. In addition, we don't need to see all the graphs for all the different countries.

I had some problems with the way the material is laid out. For example, in lines 193 and onwards it refers to hypotheses which are in Supp Materials. I think this is a mistake and the hypotheses should be upfront and not hidden away in supporting information. Similarly line 117 refers to GDPc which is explained later in the manuscript. What is Darwin Core (line 181) – may be that is explained elsewhere? Looks like this manuscript was submitted to Nature first and then passed on to Nature Communications – I suggest some consideration of expanding the main part of the article so that the reader is not asked to go between a plethora of valuable supporting documents.

Fig 1 No need for 'The' in the title

(Remarks on code availability)

NA

Reviewer #2

(Remarks to the Author)

Thank you for the opportunity to review this manuscript. While it is not always apparent to readers, the amount of data compilation and effort involved in assembling such country-level species lists—with associated metadata such as year of description and bibliographic references—is considerable. The integration of this foundation with openly available resources like GBIF, alongside comprehensive data analysis, constitutes a valuable and credible contribution to the field. I find the methodology both sound and appropriate.

The comments below are minor and relate primarily to data interpretation. These could be addressed either in the introduction or the discussion:

In ecology, the notion of a shifting baseline can obscure emerging patterns. In the context of this study, it may be worth reflecting on what the baseline actually represents. Given the evolving nature of species concepts—both historically and prospectively—can the authors comment on how taxonomic practices have changed over time (then and now, only)? Are recent discoveries more likely to involve the splitting of previously singular species/detection of cryptic taxa? A shift is certainly reflected in increasingly complex identification keys.

The Fiji *Lasioglossum* is used as an example on multiple occasions (from 4 to 13), but at the same time also serves as a reminder of the difficulties. Several of those species are known from very few specimens. There might be additional still undescribed, and they were not revised previously. Is it then really an example of how phylogeny can inform, or simply yet another group that had not previous taxonomic revisionary work conducted?

The projected 32–45 years of work needed to fully describe the fauna raises further questions: Is this projection based on a continuation of current levels of effort, or is a tapering expected as the species pool becomes more thoroughly sampled?

Have taxonomists systematically prioritized the more accessible taxa first, leaving more challenging groups for future work? Difficulties may stem from either poor sampling efficiency or a lack of distinguishing morphological/molecular characters in certain hyperdiverse clades.

In summary, I have no major concerns and am eager to cite this work and apply its methodologies to other groups, across different spatial and taxonomic scales, in support of biodiversity research.

(Remarks on code availability)

Reviewer #3

(Remarks to the Author)

In this manuscript, Dorey and colleagues present an analysis of a dataset published by some of the authors in 2023 (DOI: 10.1038/s41597-023-02626-w) combined with a taxonomical framework and country-level checklist (from the Discover Life guide, also generated by some of the authors), and an additional literature survey to find additional species missing from those data sources. Then, they used two alternative methods (both developed by Anne Chao and collaborators) to generate estimates of species richness (order 0 Hill number) – or its lower bound. By pooling all data, or binning by continent or country, global, continental or country-wise estimates are given. These estimates are then compared to the actual count of total species from the data at each of those levels, to generate what they call “taxonomic gaps”, meaning the “minimum number of species remaining to be discovered/described” at each of those levels. Then, they use linear models to try correlating the size of these gaps to several correlates like per capita GDP and continent- or country-specific traits of the dataset. The authors focused on bees (the group of hymenopteran families known as Anthophila), but their approach is straightforward to apply to any other taxon. Although the methodology for the analysis is rather scantily described in the Methods, the authors make available a set of R-based resources that allow reproduction of their results, detailed insight in their methodological approach, and re-usability for other taxa.

The authors claim that their “framework and R package toolset we provide here represents the first of its kind for any insect taxon and can be used for estimating actual taxonomic diversity across larger taxonomic and geographic scales”. They take their global estimate at face value and compare the continental and country level estimate to available checklists, discussing taxonomic gaps as a fact. Honestly, I do not think this claim can be made. Other people have used Chao’s approximation on datasets built from occurrence data to estimate richness (at regional and global scales, even for bees), or else tried using data from checklists. There is even a previous 2021 article (10.1016/j.cub.2020.10.053) in which the authors already use the same kind of data to estimate worldwide distribution patterns of bee richness.

The main issue with this analysis is that Chao’s estimates were mainly aimed at comparing diversity at the community level across samples or sites and assume a level of homogeneity that is not really fulfilled by global datasets. If we are to accept that these richness estimates are biologically valid as absolute quantities, at least some grounding evidence should be presented. However, at no point in the text is the validity of evidence ever questioned. The authors mentioned “haphazard species richness estimates mis-cited and proliferated in literature and science communication” without providing examples, but the 20-21k number that is most often thrown (even here by the authors) comes from Michener’s book (though the authors cite their own online sources). Their estimate (between ~24k and ~26k species) looks like it could be right, but in the absence of any measurement of the actual accuracy of the approach, these numbers are still (statistically informed) guesses. This is not to say there is no value in this work – in fact, data curation and literature search made here is invaluable and praise-worthy. But it is crucial to make explicit the fact that the patterns detected in the analysis are data patterns, not biological patterns. This is evidenced by the fact that correlated with taxonomic gaps were GDPc (negatively, marginally significant compared to other significant correlates), the number of species already known (positively, largest effect), the number of clean records (negatively), the proportion of species added from literature (positively), and the interaction between the last two. All these correlates are directly related to data quality/distribution, while things like elevation range, road distance or local education had no effect.

In summary, in my humble opinion, this manuscript represents a remarkable data wrangling and curation effort, combined with generation of very useful resources and tools usable way beyond the bee-based case study presented here. However, it suffers from a high stakes claim presented without independent or grounding supporting evidence: that the estimators are accurate reflections of biological richness. Unless the authors can provide a measurement of the accuracy of their estimation approach, I think they should refrain from stating that their approach yields “actual taxonomic diversity”.

(Remarks on code availability)

I did a very quick review of what is available, including the vignette and the contents of the GitHub repo. I did not try to clone or replicate the analysis locally (considering the reported runtimes) nor took the time to start digging into the code to try to figure out how exactly it was done. I found that the description given in the Methods section was not enough to figure out the

structure of the input data; I also only partially followed how they "generated" missing data out of literature-retrieved species with no occurrence data. I am quite certain the code is complete and runs correctly; however, I am under the impression that a reader should not have to get into the nitty-gritty details of the R code and scripts to figure out how results were generated.

Reviewer #4

(Remarks to the Author)

This is an important paper in providing a replicable workflow for addressing the gap in estimates of species richness at global and country scales.

I have made comments on the attached manuscript on a number of additional considerations to be made, which would strengthen the interpretation of the results, and add discussion for further studies.

(Remarks on code availability)

The vignette is extremely useful and I wish more papers included one.

Version 1:

Reviewer comments:

Reviewer #1

(Remarks to the Author)

I am happy that the authors have addressed my main concern that there was too much information in the main document. I think the manuscript is ready to be accepted

(Remarks on code availability)

Reviewer #2

(Remarks to the Author)

Dear authors and editor,

I have checked the concerns raised by myself in the first review and the responses now provided by the authors. I think they have explained and addressed all well and have no further observations.

(Remarks on code availability)

Reviewer #3

(Remarks to the Author)

After reading replies to comments made by myself and the other reviewers, I am under the impression that while they are mostly correct and acceptable, we will still get to the point of "agreeing to disagree". I believe the issue is not really about the actual work (which is laudable), but about how it is portrayed in the title and abstract... and how it will be used by the press and superficial readers. The main point in this work deals with estimating the gap between described diversity and expected diversity: it is a valid approach with important implications, as stated throughout the manuscript. However, the title and abstract stand out as selling something else, that is, the absolute number of bee species - the term "taxonomic gap" is buried in the abstract -, and most uncritical readers (that is, most people looking for something to cite when introducing their bee work) will just read the title and cite the article even if their work has nothing to do with the taxonomical gap. I believe even a subtle change in the title could make it clear what the work is really about, e.g., "How many bee species are still undescribed?..."

I have one additional criticism about the new paragraph stating hypotheses. At least within the framework of the hypothetico-deductive method, a hypothesis is a broad, testable explanation for an observed phenomenon. It proposes a potential cause-and-effect relationship or a general principle. It's the underlying reason you think something happens. It is not the expected result of a specific experiment or analysis. Yet that is exactly what is presented as a hypothesis in statements like "(w)e hypothesised that elevational range and country area would be positively correlated with our 86 response variables due to increased niche diversity and potential difficulty in sampling". I am very aware that this misuse of the term "hypothesis" is widespread... however, two wrongs (or too many, in this case) do not make a right. The hypothesis should be a tentative, testable explanation to an observed pattern, and there is a general one here related to the spatial (regional) variance in taxonomic gap across the world, and that is that current and historical differences in socioeconomic and scientific investment generate uneven rates of species description. Based on that general hypothesis, several predictions about the outcomes of specific regression analyses can be stated (similar to the presented

"hypotheses"). I encourage the authors to revise this paragraph.

(Remarks on code availability)

Reviewer #4

(Remarks to the Author)

The authors have largely addressed many of my concerns thank you.

What I think could strengthen the paper, given the many unknowns is to have a section specifically stating these e.g.

- Needing to better ascertain new findings via collections vs revisions
 - New species based on surveying unsurveyed areas vs surveyed areas
 - The number of taxonomists / outputs per taxonomist
 - Number of new species that result from molecular techniques and number that are synonymised as a result (average)
 - Phylogenetic signal
- (and some of the other comments from other reviewers) as a 'call to action'
- Needing to conduct greater censuses, ESPECIALLY in places where records were made decades ago and none since - are these species even still extant? It may be that many species are now extinct...

In the section on molecular techniques, it might be worth emphasising that such methods alone are not necessarily reliable (e.g. describing species based on a barcode alone, without morphological analysis)

I also agree with one of the other reviewers in that the authors should be more cautious in stating it yields "estimated actual taxonomic diversity". Even as estimates, I don't think you can say 'actual taxonomic diversity'; with the many unknowns, this is an estimate of taxonomic diversity. I don't know why it is 'actual' when there's no census for many countries.

I recommend minor revisions.

Sincerely,
Dr Kit Prendergast

(Remarks on code availability)

In cases where reviewers are anonymous, credit should be given to 'Anonymous Referee' and the source. The images or other third party material in this Peer Review File are included in the article's Creative Commons license, unless indicated otherwise in a credit line to the material. If material is not included in the article's Creative Commons

REVIEWER COMMENTS

Reviewer #1 (Remarks to the Author):

How many bee species are there? A quantitative global estimate

In this manuscript the authors estimate the number of bee species in the world. They estimate an 18–25% increase on the known diversity, and suggest that this will require at least 32–45 years of taxonomic research in order to describe the new species..

The authors take novel approaches to estimating bee diversity and it is helpful to have such a well-known group intensively studied in this way since it provides ways for others to replicate with other groups. However, in my view the authors have attempted to include too much in this manuscript and would be better served by separating it out into possibly three manuscripts. For example the socio-economic factors contributing to lack of taxonomic effort on bees should be in a separate paper. In addition, we don't need to see all the graphs for all the different countries.

Response: *We really appreciate the reviewer's comments here and we're glad that they see a use in our current paper. We understand the concern about the breadth of the paper and the wish to split it into multiple manuscripts. We will try to address this comment by expanding the writing of the present paper to better accommodate the Nature Communications format. We generally prefer to keep the analyses together here to tell a more-complete story, spanning from patterns to processes underlying them. A similar approach was taken with some of the authors' 2021 paper on global bee distribution and we feel that this makes the paper here a similarly complete and highly citable treatment. We think that Nature Communications is an ideal platform for such an expansive treatment. We are also happier to leave all of the per-country graphs in the supplement as we believe that this is an open and accessible way for those estimates to be accessed and assessed.*

I had some problems with the way the material is laid out. For example, in lines 193 and onwards it refers to hypotheses which are in Supp Materials. I think this is a mistake and the hypotheses should be upfront and not hidden away in supporting information.

Response: *We agree with the reviewer and are taking advantage of the increased manuscript length to do just that.*

Similarly line 117 refers to GDPc which is explained later in the manuscript.

Response: *Many thanks for picking up that error, it is now defined at first use.*

What is Darwin Core (line 181) – may be that is explained elsewhere?

Response: *Good point, this is maybe not common knowledge. We have now briefly described it and provided a link to the data standard website.*

*“... aggregate (e.g., using **the Darwin Core occurrence data standard — dwc.tdwg.org**) would ...”*

Looks like this manuscript was submitted to Nature first and then passed on to Nature Communications – I suggest some consideration of expanding the main part

of the article so that the reader is not asked to go between a plethora of valuable supporting documents.

Response: *We thank the reviewer for this comment and note that we have now made efforts to do just that in response to your review.*

Fig 1 No need for 'The' in the title

Response: *This has now been removed.*

Reviewer #1 (Remarks on code availability):

NA

Reviewer #2 (Remarks to the Author):

Thank you for the opportunity to review this manuscript. While it is not always apparent to readers, the amount of data compilation and effort involved in assembling such country-level species lists—with associated metadata such as year of description and bibliographic references—is considerable. The integration of this foundation with openly available resources like GBIF, alongside comprehensive data analysis, constitutes a valuable and credible contribution to the field. I find the methodology both sound and appropriate.

Response: *Many thanks and much appreciated. Many hands make light work!*

The comments below are minor and relate primarily to data interpretation. These could be addressed either in the introduction or the discussion:

In ecology, the notion of a shifting baseline can obscure emerging patterns. In the context of this study, it may be worth reflecting on what the baseline actually represents. Given the evolving nature of species concepts—both historically and prospectively—can the authors comment on how taxonomic practices have changed over time (then and now, only)? Are recent discoveries more likely to involve the splitting of previously singular species/detection of cryptic taxa? A shift is certainly reflected in increasingly complex identification keys.

Response: *This is a complex question and not one we can fully interrogate in the current paper, although we have made efforts to give some increased explanation. Essentially, the type of taxonomic work left to be done depends on the area, in line with our findings on underdescription. For those areas with more species left to describe, typically even the most basic accounting and taxonomic descriptions of relatively obvious species will make up a substantial portion of the necessary work. However, even the best studied countries like North America retain hundreds of species to be described in places like the species-rich SW USA deserts. In many cases, these remaining species belong to large groups that are more complex to revise, which inhibits quick description because such groups are best treated in total with comprehensive descriptions and keys.*

In regards to Africa, as an example where much work is to be done, we write: "In such cases, it seems likely that even initial efforts at taxonomic description and

revision in the region will uncover many new species and greatly enhance our ability to work with and understand the fauna.”

Conversely, we explain the situation for better known regions as “Even in the best studied regions such as North America, many species may persist undescribed in larger groups (such as *Andrena* or *Osmia* in the USA) that are best treated altogether to enhance identifiability and ensure accurate taxonomic actions.”

We then return later when discussing overall drivers to reinforce the overarching situation: “There are opportunities for foundational and traditional taxonomy in areas such as Africa to more coordinated efforts on larger or more challenging groups in places such as North America. For the former, integrative systematics will build strong taxonomic scaffolds early and for the latter it will allow robust testing of existing species hypotheses.”

We also add more nuance and raise the possibility of empirically testing this issue “Additionally, sequencing efforts can, and do, result in the discovery of synonyms (rejections of existing species hypotheses)³³. Large barcoding efforts may elucidate, by region, the extent to which existing taxonomies have over- or under-described species.”

Finally, our valid name accumulation curve actually represents the number of species **that remain valid today**. And so, at least since 1960, the amount of “good” names described per year has remained stable. As for the changes in rate prior to that further research might uncover the degree to which the number of active taxonomists and the taxonomic gaps might have contributed.

The Fiji *Lasioglossum* is used as an example on multiple occasions (from 4 to 13), but at the same time also serves as a reminder of the difficulties. Several of those species are known from very few specimens. There might be additional still undescribed, and they were not revised previously. Is it then really an example of how phylogeny can inform, or simply yet another group that had not previous taxonomic revisionary work conducted?

Response: *This example was used only once in the manuscript. However, I see the reviewer’s point that some context is missing here. We have added in that the group is morphologically cryptic. Additionally, the initial descriptions around the 1920’s included only two species, and the subsequent revision around the 1970’s included four species, but we know that samples from several more species were included under the current name “*Lasioglossum fijiense*”. Hence, these went undescribed using purely morphological analyses.*

“In a **recent** and contrasting example, the **morphologically cryptic** Fijian *Lasioglossum* bees went from four morphologically described species to 13 when systematic techniques were included²⁸, and recent molecular evidence suggests at least 28 total species²⁹; a 3.3- and 7-fold increase, respectively”

The projected 32–45 years of work needed to fully describe the fauna raises further questions: Is this projection based on a continuation of current levels of effort, or is a tapering expected as the species pool becomes more thoroughly sampled?

Response: *This is a wonderful question that we would like a good answer to ourselves. We state that this is at the current rate of descriptions, but also already note that we expect a tapering of discovery. Again, this is a fascinating line of questioning, but we don’t see a robust way to test that at present, especially considering that the rate of description has been stable for some time.*

“Applying this rate directly we might expect to close the taxonomic gap in 32–45 years. However, given that our richness estimates are a lower bound, the integration of phylogenetic analyses can be very limited (Table S1), research funding and capacity is geographically biased³⁷, reduced support for taxonomy, and **the likely diminishing returns of species revision towards the tail end of discovery, we expect this estimate to be quite conservative.**”

Have taxonomists systematically prioritized the more accessible taxa first, leaving more challenging groups for future work? Difficulties may stem from either poor sampling efficiency or a lack of distinguishing morphological/molecular characters in certain hyperdiverse clades.

***Response:** This is a good question. We don't have a universal response for that, as taxonomic discovery and description are a process at different stages depending on the region. We can point out that, in Australia, some of the more-recent taxonomic works, some very difficult groups have been tackled — several revisions of *Lasioglossum* groups for example. But also a recent revision of *Megachile* (*Austrochile*) which could be considered either easy or difficult depending on how you look at it. *Bombus* is a great example of a group that might be assumed to be easy but in fact is very challenging due to mimicry etc. *Anthophorini* could be an example of a charismatic and long-studied group that is actually very difficult and unresolved.*

We can't comment too much and aren't sure how to get at that question easily or at a global scale, as this requires different methods beyond our scope and possibly even expert opinion surveys, but we hope that the added text from above where discussing the process helps to suffice and explain the current state of things (especially in regards to many new species clustered in bigger, tougher groups).

In summary, I have no major concerns and am eager to cite this work and apply its methodologies to other groups, across different spatial and taxonomic scales, in support of biodiversity research.

***Response:** Once again, many thanks for your positive and thoughtful comments. There is room to think more and look for ways to answer more questions about this area of research.*

Reviewer #3 (Remarks to the Author):

In this manuscript, Dorey and colleagues present an analysis of a dataset published by some of the authors in 2023 (DOI: 10.1038/s41597-023-02626-w) combined with a taxonomical framework and country-level checklist (from the Discover Life guide, also generated by some of the authors), and an additional literature survey to find additional species missing from those data sources. Then, they used two alternative methods (both developed by Anne Chao and collaborators) to generate estimates of species richness (order 0 Hill number) – or its lower bound. By pooling all data, or binning by continent or country, global, continental or country-wise estimates are given. These estimates are then compared to the actual count of total species from the data at each of those levels, to generate what they call “taxonomic gaps”, meaning the “minimum number of species remaining to be discovered/described” at each of those levels. Then, they use linear models to try correlating the size of these

gaps to several correlates like per capita GDP and continent- or country-specific traits of the dataset. The authors focused on bees (the group of hymenopteran families known as Anthophila), but their approach is straightforward to apply to any other taxon. Although the methodology for the analysis is rather scantily described in the Methods, the authors make available a set of R-based resources that allow reproduction of their results, detailed insight in their methodological approach, and re-usability for other taxa.

Response: *Thank you for your constructive feedback. We do describe our methods relatively concisely, although we hope with enough information to be reproducible. However, our results are truly reproducible through making all of our code and data publicly available on GitHub. We are happy to add additional detail wherever it is suggested as necessary.*

The authors claim that their “framework and R package toolset we provide here represents the first of its kind for any insect taxon and can be used for estimating actual taxonomic diversity across larger taxonomic and geographic scales”. They take their global estimate at face value and compare the continental and country level estimate to available checklists, discussing taxonomic gaps as a fact. Honestly, I do not think this claim can be made. Other people have used Chao’s approximation on datasets built from occurrence data to estimate richness (at regional and global scales, even for bees), or else tried using data from checklists. There is even a previous 2021 article (10.1016/j.cub.2020.10.053) in which the authors already use the same kind of data to estimate worldwide distribution patterns of bee richness. The main issue with this analysis is that Chao’s estimates were mainly aimed at comparing diversity at the community level across samples or sites and assume a level of homogeneity that is not really fulfilled by global datasets.

Response: *We think that this is an interesting point to raise and it does have some merit. However, we don’t agree that the Chao’s estimates, particularly *iChao*, has such a narrow definition. For example, Chao described statistical estimates of “richness” very broadly to include “They may be biological species, bugs in software programs, words in a book, genes or alleles in genetic code, or other discrete entities”. Additionally, *iChao* is a non-parametric estimator in the sense that “In this article, we focus on a nonparametric approach in the sense that no assumptions are made about the underlying distribution of species abundances”.*

*Heterogeneity — in species-level abundances — can indeed negatively bias Chao’s estimators. However, this is likely to be the case in any community. Additionally, see: “In the **homogeneous** model (Figure 1a and Web Table C1), the Chao1 and the *iChao1* estimators are close to each other (the latter has slightly higher positive bias), and both are nearly unbiased. **For heterogeneous models (CVformula0), both estimators have negative biases because they are derived as lower bounds.** Nevertheless, they follow the intuitive pattern: The magnitude of bias and RMSE decrease as sample size increases. The *iChao1* estimator has smaller bias (Figure 1), but larger variance (in all tables) than the Chao1 estimator due to its use of more frequencies to estimate the number of undetected species. **In terms of bias, the proposed *iChao1* estimator always outperforms the Chao1 estimator (Figure 1).**” — <https://doi.org/10.1111/biom.12200>*

*In comparison, hill numbers (*iNEXT*), “When $q = 0$, *0D* is simply species richness, which counts species **equally without regard to their relative abundances.**” — <https://doi.org/10.1111/2041-210X.12613>*

I have added “The iChao estimate is known to perform well when species abundances in “communities” are homogenous and can be negatively biased where they are heterogenous⁶⁹. Hill numbers, as estimated in iNEXT, are implemented to make fairer comparisons between assemblages that may vary in size^{59,71}.” To the methods and have added another reference (71). We think that our set of estimators together provides a relatively less biased and highly reliable estimate of bee species richness at the global scale.

Notably, we should also add that the 2021 paper by some of the authors was unable to use absolute richness, only relative expected species richness. This paper is a very big step forward from the methods of the prior one, in terms of both the data available and the efforts to ascertain true species richness values for bees globally.

If we are to accept that these richness estimates are biologically valid as absolute quantities, at least some grounding evidence should be presented. However, at no point in the text is the validity of evidence ever questioned.

Response: *We argue that we do question the validity of our evidence throughout the text. However, we also add further caveats to try and make our uncertainties clearer. For example:*

*“Our estimates provide a powerful guide **and useful hypotheses** for where to concentrate species discovery and revisionary research”*

*“However, for Africa, increasing GDPc appeared to be associated with an increase in the number of undescribed species (Fig. S11A). **This likely reflects an overall larger taxonomic gap for Africa and an inability to reliably estimate trends due to a lack of data** (Figs S11F-G).”*

*“Without log-transformation we maintained our hypotheses that the number of clean occurrence records and the proportion of occurrence-based species were negatively associated with our response variables because both variables indicate greater sample completeness associated with more taxonomic work and material (Supp. Materials). **This makes intuitive sense, as areas with more data should have undergone more taxonomic research and identification work necessary to generate such data.**”*

*“To determine the potential **impact of sampling species counts from the empirically derived literature curve, we further sampled the two curves based on the empirical country- and global-level occurrence datasets. Most estimates were robust to changes in the chosen curve, with the country-sampled occurrence curve sometimes producing higher estimates than the literature or global curves** (Figs. S10 and S13). This suggests that our use of the empirical literature curve is a conservative and a robust choice and hence useful for implementation with other taxa.”*

*“**Verification of our results could also be achieved by surveying the number of undescribed species in museums.** However, this would require a colossal digitization effort which has not even been adequately done for described materials.”*

*“The relatively low estimate for Oceania was unanticipated. **However, we caution that this might be driven by relatively high data availability in Australia, but poor regional availability,** and insufficient use of molecular techniques even in recent taxonomic research. Insufficient implementation of molecular techniques in taxonomy is an issue for many countries and taxa.”*

*“**We applied several tests of the data and found that the significant patterns and directions of fixed effects in our model were mostly robust to***

changes in the models; however, changing the transformations could alter the results. Importantly, without log-transforming the number of clean records and the proportion of occurrence-based species the significance values were identical but both fixed effects variables were negatively associated with the response variables. We additionally sampled curves derived from the global- and country-level species occurrence data (Fig. S10). Except for GDPc, and — for the global dataset — the interaction term between the number of clean records and the proportion of occurrence-based species, the significance and sign of the remaining fixed effects were consistent (Table S3)."

"Furthermore, and in order to examine the interaction between terms, **we extracted data from a curve based off of the (a) global and (b) per-country species occurrence data frequencies as we did with the literature data** (Fig. S10). These data were normalized with the literature curve to work from the same scale during formula production and plotting. **We ran these analyses ten times at the country level for iChao and iNEXT to examine how changing empirical data that fills in the no-occurrence species impacts the diversity gap analyses.**"

The authors mentioned "haphazard species richness estimates mis-cited and proliferated in literature and science communication" without providing examples, but the 20-21k number that is most often thrown (even here by the authors) comes from Michener's book (though the authors cite their own online sources). Their estimate (between ~24k and ~26k species) looks like it could be right, but in the absence of any measurement of the actual accuracy of the approach, these numbers are still (statistically informed) guesses.

Response: *This is a good comment and we have added reference to Michener's book in our manuscript "In 2007, Michener²³ estimated that there were ~18,000 described species with >20,000 total in the world.". We think that this improves the manuscript and the context.*

However, Michener estimated that there were 20k+ species in the world and also references conversations with John Ascher (an included author) and on an early draft of the Discover Life checklist in coming up with these estimates. The numbers cited in the manuscript actually refer to the current empirical number of described bee species in the world. We still argue that this is a haphazard estimate "A guess as to the total number of bee species in the world is therefore near or above the often-mentioned figure of 20,000." — Michener 2007

This is not to say there is no value in this work – in fact, data curation and literature search made here is invaluable and praise-worthy. But it is crucial to make explicit the fact that the patterns detected in the analysis are data patterns, not biological patterns. This is evidenced by the fact that correlated with taxonomic gaps were GDPc (negatively, marginally significant compared to other significant correlates), the number of species already known (positively, largest effect), the number of clean records (negatively), the proportion of species added from literature (positively), and the interaction between the last two. All these correlates are directly related to data quality/distribution, while things like elevation range, road distance or local education had no effect.

In summary, in my humble opinion, this manuscript represents a remarkable data wrangling and curation effort, combined with generation of very useful resources and tools usable way beyond the bee-based case study presented here. However, it suffers from a high stakes claim presented without independent or grounding

supporting evidence: that the estimators are accurate reflections of biological richness. Unless the authors can provide a measurement of the accuracy of their estimation approach, I think they should refrain from stating that their approach yields “actual taxonomic diversity”.

Response: *We understand the reviewers concern here. While we also point out that we state that we are “**estimating** actual taxonomic diversity”.*

Sadly, no census for bee species richness exists for any country. However, we did have some assumptions about countries that might asymptote prior to beginning our analyses like Sweden and Singapore and these countries did indeed [nearly] asymptote. We did not formalise these hypotheses and so did not consider it fair to present as evidence.

We do additionally provide some tests of the impact of how we generate the literature curve and believe that we have made a conservative estimate. Additionally, from estimates made by Chao et al. we believe that we have conservative estimates of species richness

(https://figshare.com/articles/dataset/Appendix_F_Using_simulation_to_test_the_proposed_analytic_estimators_/3568410?backTo=%2Fcollections%2FRarefaction_and_extrapolation_with_Hill_numbers_a_framework_for_sampling_and_estimation_in_species_diversity_studies%2F3309822&file=5645115).

I also feel that similar concerns could be raised around most large-scale ecological modelling techniques. But, we maintain that our estimates are more reliable than any haphazardly-derived estimates to date and they provide estimates of error around which to test. However, we do want to make this clear and so have added the following paragraph:

“Such analyses should be made and explained with caveats kept in mind. For example, while iChao and Hill numbers are relatively robust estimators of species richness within assemblages⁵⁵⁻⁵⁷, our “assemblages” are large. Global species occurrence datasets, while relatively extensive, have inherent biases²⁸; however, biases abound for perhaps all sampling methods⁵⁸. We expect that taxonomic concepts will also impact our estimates, particularly where rare or cryptic species are left undescribed or lumped. While we state species richness estimates here at several levels, we highlight that they are estimates of the lower bound of diversity (biases are likely to further decrease estimates⁵⁵⁻⁵⁷) and that they can be somewhat sensitive to input data (e.g., Figs S10 and S13). We suggest that estimates, while better than haphazard estimates, are appropriately used as statistically derived hypotheses and roadmaps for future work and management.”

We hope that you find the added explanation and caveats sufficient to make clear the potential limits of the study and the need for future work to provide validations of our estimates provided here.

Reviewer #3 (Remarks on code availability):

I did a very quick review of what is available, including the vignette and the contents of the GitHub repo. I did not try to clone or replicate the analysis locally (considering the reported runtimes) nor took the time to start digging into the code to try to figure out how exactly it was done. I found that the description given in the Methods section was not enough to figure out the structure of the input data; I also only partially followed how they "generated" missing data out of literature-retrieved species with no occurrence data. I am quite certain the code is complete and runs correctly;

however, I am under the impression that a reader should not have to get into the nitty-gritty details of the R code and scripts to figure out how results were generated.

Response: *We will undertake a check of the methods to see if things can be made clearer and if the editor believes further clarification is required in the available space, we will seek to further simplify. We do agree that readers should not necessarily need to understand the R code to understand the analysis. However, R code of course provides an opportunity for any reader to get into the nitty-gritty and see exactly what we have done. Both our annotated workflow and our vignette are intended to serve this purpose as a supplement to understanding what's presented in the manuscript.*

Reviewer #4 (Remarks to the Author):

This is an important paper in providing a replicable workflow for addressing the gap in estimates of species richness at global and country scales.

Response: *We greatly appreciate your supportive words and hope that the manuscript will have a positive impact more broadly.*

I have made comments on the attached manuscript on a number of additional considerations to be made, which would strengthen the interpretation of the results, and add discussion for further studies.

Notes from the annotated document:

"I think it might be prudent to look at a phylogenetic signal" ... also notes on the number of taxonomists and undescribed specimens in museums.

We appreciate this comment and we had considered attempting to include phylogenetic information. However, we consider the existing data on bees to be insufficient to make many meaningful assertions. An alternative might be to run these analyses at higher-level groupings like family, subfamily, or tribe. However, we considered that this would take an already massive manuscript and blow it up too far. It could be a good topic for a further manuscript.

We also agree that correlating the number of taxonomists and undescribed species in museums would be a fantastic way to attempt to examine our results. However, many problems arise, particularly in compiling those data (especially the latter would be impossible without a huge funding and digitization effort for each country). Similarly, people often do different quantities of taxonomic work at different stages in their career these days, depending on the career expectations and other considerations in their fields and areas. Regardless, we have noted all of these in the manuscript because we do agree with your assertions, and perhaps there are taxa for which those data will be easier to compile.

"But the Austrochile just had 71 new species described ... with previously 1700 described species, and generally estimates by taxonomists eg Terry Houston there's at least 500 undescribed species, that suggests a taxonomic gap of about 30%"

Response: *Yes, you are absolutely correct and we have been unsure if we should include these new papers in a revision of the work. The first author is also a co-author on that work but the timing of publications was not ideal. However, he is hoping to analyse those data separately and undertake a deeper dive into the*

Australian data using datasets like that made available by Remko et al. Notably, Australia was also a major challenge for earlier efforts at mapping bee species richness (Orr et al. 2021), given the unusual colletid-heavy fauna and the distribution of most data records along the coast rather than the interior. As such, it will be useful to perform additional studies specifically on Australia.

“Interesting to see the relative contribution of new species based on revisions of a group, vs new species based on new collection records etc.”

Response: *That is a very interesting idea. There are some new R packages in development to digitise collection data from taxonomic works (amazing!). Such a package will make answering such a question feasible — the lead author started to digitise records from taxonomic works and the time input was too considerable to do this at scale. We suspect that this could be done soon though and we’d be interested in trying to help out.*

“Is there a way to quantify this? E.g. number of authors on bee taxonomic publications per country over the last 20 years?”

Response: *Another very good question. We had been getting at taxonomic capacity through sociopolitical data and this would require building a new dataset of all of the people who have worked on bees and where they were from... or if they were from multiple countries... maybe where they have worked throughout their life. We thought that this was interesting but another large and detective-intensive dataset to collate. But, again, we think that this could be an interesting exercise! We do at least have the last names of all bee taxonomists and the years that they were active! Then you have to separate the Schwarz’s and the Schwarz’s, etc. Additionally, we’d need to think about the probable increase in the number of co-authors on more recent taxonomic works and how to measure those. For the present manuscript, we feel like this is outside of the scope, given that we use multiple metrics that should correlate relatively well with numerical taxonomic capacity.*

...The remaining minor line comments have all been integrated in the new manuscript

Reviewer #4 (Remarks on code availability):

The vignette is extremely useful and I wish more papers included one.

Response: *We greatly appreciate your kind comments in this regard. We agree with you in this instance that the best way to have method actually implemented by the research community is to remove as many barriers to use as possible.*

REVIEWERS' COMMENTS

Reviewer #1 (Remarks to the Author):

I am happy that the authors have addressed my main concern that there was too much information in the main document. I think the manuscript is ready to be accepted

Response: We greatly appreciate your efforts in improving our work!

Reviewer #2 (Remarks to the Author):

Dear authors and editor,

I have checked the concerns raised by myself in the first review and the responses now provided by the authors. I think they have explained and addressed all well and have no further observations.

Response: We thank you for your effort and your help!

Reviewer #3 (Remarks to the Author):

After reading replies to comments made by myself and the other reviewers, I am under the impression that while they are mostly correct and acceptable, we will still get to the point of "agreeing to disagree". I believe the issue is not really about the actual work (which is laudable), but about how it is portrayed in the title and abstract... and how it will be used by the press and superficial readers. The main point in this work deals with estimating the gap between described diversity and expected diversity: it is a valid approach with important implications, as stated throughout the manuscript.

However, the title and abstract stand out as selling something else, that is, the absolute number of bee species - **the term "taxonomic gap" is buried in the abstract** -, and most uncritical readers (that is, most people looking for something to cite when introducing their bee work) will just read the title and cite the article even if their work has nothing to do with the taxonomical gap. I believe even a subtle change in the title could make it clear what the work is really about, e.g., "**How many bee species are still undescribed?...**".

Response: We have adopted the title suggested by the journal

I have one additional criticism about the new paragraph stating hypotheses. At least within the framework of the hypothetico-deductive method, a hypothesis is a broad, testable explanation for an observed phenomenon. It proposes a potential cause-and-effect relationship or a general principle. It's the underlying reason you think something happens. **It is not the expected result of a specific experiment or analysis.** Yet that is exactly what is presented as a hypothesis in statements like "**(w)e hypothesised that elevational range and country area would be positively correlated with our response variables due to increased niche diversity and potential difficulty in sampling**". I am very aware that this misuse of the term "hypothesis" is widespread... however, two wrongs (or too many, in this

case) do not make a right. The hypothesis should be a tentative, testable explanation to an observed pattern, and there is a general one here related to the spatial (regional) variance in taxonomic gap across the world, and that is that current and historical differences in socioeconomic and scientific investment generate uneven rates of species description. Based on that general hypothesis, several predictions about the outcomes of specific regression analyses can be stated (similar to the presented "hypotheses"). I encourage the authors to revise this paragraph.

Response: We have revised this paragraph and believe that it reads better now. The paragraph is also in line with the reviewer's suggestions.

"We hypothesize that contemporary and historical differences in socioeconomic and scientific investment as well as physical regional characteristics have generated uneven rates of bee species description across regions. We propose several predictions in relation to our response variables, predicted number and percentage of new species, to address this hypothesis. Firstly, due to increased niche diversity and potential difficulty in sampling, we predict that our response variables will be positively correlated with elevational range and country area. Secondly, because increased median distance to nearest road could make sampling for bees more difficult, our response variables will be positively correlated with median distance to nearest road. Thirdly, we predict that our response variables will be positively correlated with observed species richness. This is because we expect that the taxonomic load might be higher; however, we also recognized that it could indicate that a larger portion of the bee diversity was already described. Fourthly, we predict that our response variables will be negatively correlated with gross domestic capital per capita (GDPc) and education. This is because we expected that GDPc and education would be indicators of a country's investment into research and a greater capacity to undertake taxonomic works locally or to host collaborating taxonomists. Finally, we predicted that, due to greater inventory and taxonomic efforts, our response variables would be negatively correlated with the number of clean occurrence records and the proportion of occurrence-based species per country."

Reviewer #4 (Remarks to the Author):

The authors have largely addressed many of my concerns thank you.

What I think could strengthen the paper, given the many unknowns is to have a section specifically stating these e.g.

- Needing to better ascertain new findings via **collections vs revisions**

*Response: I'm not certain exactly what the reviewer wants here. However, in relation to collections vs revisions, we think that these should go hand-in-hand. We also believe that this will be regionally specific but that most places with a taxonomic gap likely require both more collection effort and more revisionary works. Regardless, we have added some of this context here "However, for Africa, increasing GDPc appeared to be associated with an increase in the number of undescribed species (Fig. S11A). This likely reflects an overall larger taxonomic gap for Africa and an inability to reliably estimate trends due to a lack of data (Figs S11F-G). In such cases, it seems likely that even initial efforts at **collections and***

taxonomic research in the region will uncover many new species and greatly enhance our ability to work with and understand the fauna.”

- New species based on surveying unsurveyed areas vs surveyed areas

Response: This is perhaps too specific a recommendation to give at a global, continental, or country level. We highlight the country level regions that require more work, but most countries are large and have diverse regions that are sampled better or worse. Other works highlight at a more granular level where we might expect more diversity to be found at a relative level (e.g., Orr et al 2021), even if they don't make quantitative estimates. Our workflow could be used very easily at lower spatial resolutions by treating Bioregions, Biomes, Vegetations classifications, etc. as habitats for which to estimate species richness. But, it's too granular for this specific work. We think that this work should certainly be used to prioritise those granular efforts and that those efforts likely would strongly benefit from regional expertise to interpret.

- The number of taxonomists / outputs per taxonomist

Response: As in the last revision this isn't as simple a project as you might think. Particularly as authors with the same name need to be split per species. That would represent a whole new digitisation and quality assurance phase. This requires a very specific type of dataset on species, their types, and their authors, that we did not have access to for this paper. I think that interesting work could be done based around this idea as a whole, rather than doing a poor job of it in this manuscript when it is tangentially related. I am also told that such an effort might already be underway.

- Number of new species that result from molecular techniques and number that are synonymised as a result (average)

Response: This would again represent a major review process and would have to form its own work. To be presented in this work would require a global review of all bee taxonomy papers. This should be done systematically if it's to be done.

- Phylogenetic signal

(and some of the other comments from other reviewers) as a 'call to action'

Response: We're not sure what the reviewer is suggesting here. Phylogenetic signal — as the tendency of closely related species to resemble one another — doesn't immediately seem to fit into our work. I agree that using phylogenetics at some large scale would be interesting. A large global bee phylogeny exists, but this matrix-based phylogeny is not well-enough resolved to be of global use. We also think that such a work should be done with great care and attention to detail and would be worthy of its own hypothesis-driven science. Our present statistics are, by-and-large, taxon-agnostic.

- Needing to conduct greater censuses, ESPECIALLY in places where records were made decades ago and none since - are these species even still extant? It may be that many species are now extinct...

Response: We hope that our work leads to great taking of census information and targeted research. To make broad claims at a global level would be too much for this current paper — which has already been called too large for a single work by

one of the reviewers. We appreciate your enthusiasm and believe that there are good ideas here. For us, we would like to see these ideas addressed fairly, with focussed and data-driven research.

In the section on molecular techniques, it might be worth emphasising that such methods alone are not necessarily reliable (e.g. describing species based on a barcode alone, without morphological analysis)

*Response: See the section with “Hence, we should expect that these estimates will increase overall as molecular sampling techniques are implemented more-broadly, sequencing costs fall, and data availability increases³¹. However, it is important that these techniques are **used reliably and without over-interpretation**^{31,32}.” We do not advocate for barcode-only taxonomy. We strongly advocate for a robust and systematic approach to taxonomy. See also “This, in combination with a lack of recent and thorough integrative **systematic** research in Oceania, could explain the lower estimates where we otherwise expected higher richness. A concurrently published paper, not included in our analyses, highlighted this issue by describing 71 new species of Australian *Megachile* (*Austrochile*) (28% of the estimated taxonomic gap of 250 species) **using systematic techniques**⁴⁰.”*

I also agree with one of the other reviewers in that the authors should be more cautious in stating it yields “estimated actual taxonomic diversity”. Even as estimates, I don't think you can say 'actual taxonomic diversity'; with the many unknowns, this is an estimate of taxonomic diversity. I don't know why it is 'actual' when there's no census for many countries.

Response: That line has already been removed. Perhaps the issue is with the word “Actual” in this instance? We can see that this might strengthen the sentence and its meaning and so we were happy to remove it.

I recommend minor revisions.

Sincerely,
Dr Kit Prendergast